# Cullin 3 Exon 9 Deletion in Familial Hyperkalemic Hypertension Impairs Cullin3-Ring-E3 Ligase (CRL3) Dynamic Regulation and Cycling

**DOI:** 10.3390/ijms23095151

**Published:** 2022-05-05

**Authors:** Ilektra Kouranti, Waed Abdel Khalek, Stephani Mazurkiewicz, Irmine Loisel-Ferreira, Alexis M. Gautreau, Lionel Pintard, Xavier Jeunemaitre, Eric Clauser

**Affiliations:** 1Université Paris Cité, French National Institute of Health and Medical Research (INSERM), Paris Cardiovascular Research Center (PARCC), F-75015 Paris, France; ilektra.kouranti@gmail.com (I.K.); waed.ak@gmail.com (W.A.-K.); stefmazurkiewicz@gmail.com (S.M.); irmine.ferreira@inserm.fr (I.L.-F.); xavier.jeunemaitre@inserm.fr (X.J.); 2Laboratory of Structural Biology of the Cell, Ecole Polytechnique, Institut Polytechnique de Paris, 91120 Palaiseau, France; alexis.gautreau@polytechnique.edu; 3Center of Molecular and Cellular Biology (CMCB), Skolkovo Institute of Science and Technology (Skoltech), 12105 Moscow, Russia; 4French National Centre for Scientific Research (CNRS), Institut Jacques Monod, Université Paris Cité, F-75013 Paris, France; lionel.pintard@ijm.fr

**Keywords:** WNK kinase, BTB protein, Cullin, CRL complex, interactome

## Abstract

Cullin 3 (CUL3) is the scaffold of Cullin3 Ring E3-ligases (CRL3s), which use various BTB-adaptor proteins to ubiquitinate numerous substrates targeting their proteasomal degradation. *CUL3* mutations, responsible for a severe form of familial hyperkalemia and hypertension (FHHt), all result in a deletion of exon 9 (amino-acids 403-459) (CUL3-∆9). Surprisingly, while CUL3-∆9 is hyperneddylated, a post-translational modification that typically activates CRL complexes, it is unable to ubiquitinate its substrates. In order to understand the mechanisms behind this loss-of function, we performed comparative label-free quantitative analyses of CUL3 and CUL3-∆9 interactome by mass spectrometry. It was observed that CUL3-∆9 interactions with COP9 and CAND1, both involved in CRL3 complexes’ dynamic assembly, were disrupted. These defects result in a reduction in the dynamic cycling of the CRL3 complexes, making the CRL3-∆9 complex an inactive BTB-adaptor trap, as demonstrated by SILAC experiments. Collectively, the data indicated that the hyperneddylated CUL3-∆9 protein is inactive as a consequence of several structural changes disrupting its dynamic interactions with key regulatory partners.

## 1. Introduction

Familial hyperkalemic hypertension (FHHt), also known as Gordon syndrome or Pseudohypoaldosteronism type 2 (OMIM #145260), is a rare monogenic disease characterized by high blood pressure, hyperkalemia, and hyperchloremic acidosis. The first mutations identified in FHHt were in genes expressing the WNK1 and WNK4 serine/threonine kinases, which are abundantly expressed at the distal convoluted tubule of the nephron. These kinases activate downstream kinases (SPAK, OSR1), which in turn phosphorylate and activate the Na^+^-Cl^−^ co-transporter (NCC), leading to NaCl reabsorption. These mutations result in accumulation of the WNK1 or WNK4 kinase leading to hypertension and metabolic disorders. 

Subsequently, mutations in *CUL3* and *KLHL3 (Kelch like family member 3),* two genes encoding components of a cullin 3-RING-E3 ligase complex (CRL3), have been shown to cause FHHt [1,2].

Cullin 3 (CUL3) is the scaffold protein of the CRL3 complex, which ubiquitinates numerous substrates for proteasomal degradation. The substrates interact with the N-terminal part of CUL3 via its BTB/POZ domain [3]. The C-terminal part of CUL3 interacts with E2 ubiquitin enzymes via the RING finger protein RBX1. 

In order to rapidly adapt the ubiquitination machinery to the available substrates, CRL complexes cycle between active and inactive states in order to exchange adaptor subunits and rapidly adapt the CRL repertoire to the available substrates to be degraded. In the inactive state, the CUL3-RBX1 catalytic core is associated with the exchange factor Cullin-Associated and Neddylation Dissociated protein 1 (CAND1), which prevents CUL3 binding to any adaptor and substrate. In the first step of activation, CAND1 catalyzes the binding of an adaptor and then dissociates from CUL3. Subsequently, CUL3 is neddylated, the adaptor binds the substrate, and RBX1 associates to the ubiquitin-carrying enzyme, allowing mono- or polyubiquitination of the substrate and its proteasomal degradation. When the polyubiquitinated substrate dissociates from CRL3, the C-terminal domain of CUL3 binds to the COP9 signalosome (CSN), which hydrolyzes Nedd8 conjugates [4,5]. Finally, adaptor dissociation allows re-association of the CUL3-RBX1 catalytic core to CAND1.

Previous work established that KLHL3 acts as an adaptor subunit of a CRL3 complex targeting WNK1 and WNK4 kinases for degradation, thereby linking CRL3^KLHL3^ to FHHt. Most *WNK1, WNK4,* and *KLHL3* mutations in FHHt impair interactions of WNK or CUL3 with the adaptor KLHL3 and result in impaired degradation (i.e., accumulation) of WNKs [6]. More intriguing are the FHHt-related CUL3 mutations, which all result in skipping of exon 9, producing an in-frame fusion of exons 8 and 10 [1,7]. Patients carrying those mutations present a more severe form of FHHt, with earlier onset, higher blood pressure, and more severe metabolic disorders than the patients with mutations affecting the three other genes. Several groups have tried to elucidate the molecular and cellular functions of the CUL3 mutant, with interesting but sometimes contradictory conclusions [8,9,10], including spurious degradation of CUL3 adaptors instead of substrates, increased flexibility of the C-terminal domain of CUL3, or modifications of CUL3-∆9 interactions with several regulatory proteins. 

To elucidate the molecular basis of FHHt-related CUL3 mutations, we performed a comprehensive and comparative proteomic analysis of CUL3 versus CUL3-∆9 in human cells and employed SILAC experiments to explore the dynamic assembly of wild-type and mutated CRL3 complexes. Our results highlight multiple levels of inhibition caused by the CUL3-∆9 mutant that likely co-exist in several tissues, contributing to a severe form of FHHt.

## 2. Results

### 2.1. Interactome Comparison of CUL3-WT and CUL3-∆9 in Stable and Inducible Cell Lines

To investigate the potential cause of CUL3-∆9 phenotypes, we performed quantitative mass spectrometric analyses of wild-type CUL3 (CUL3-WT) and CUL3 deleted for exon 9 (CUL3-∆9) complexes. To do so, we generated stable cell lines expressing N-terminally tagged CUL3 or CUL3-∆9 with 6xHis and a Protein C tag (PrC) under the control of the inducible Tet promoter in the 293 Flp-In T-Rex cell line (Figure 1A).

As previously described [9], PrC-CUL3-∆9 was much more heavily neddylated (59% ± 2%) compared to PrC-CUL3-WT (11.6% ± 2.5%; *n* = 13, *p* < 0.0001) in those cell lines (Figure 1B). 

Additionally, PrC-CUL3-∆9 was less abundant than PrC-CUL3-WT, although both were integrated at the same genetic locus (see the Materials and Methods section). This difference in protein amounts was quantified on numerous Western blots (38% ± 10%, *n* = 14, *p* < 0.01) (Appendix A). We also explored the potentially accelerated degradation of CUL3-∆9, which has been suggested by several authors to explain the loss-of-function of this mutant. Using a cycloheximide chase assay on bioluminescent-tagged CUL3-WT or CUL3-∆9 proteins, no difference in CUL3-WT versus CUL3-∆9 half-life was observed (Appendix A). In addition, the amount of the CUL3 mRNA was not significantly different in either cell line (Appendix A), and their degradation rates was similar (Appendix A). In the absence of differences in amounts of mRNA and in degradation rates of mRNA and protein, it can be speculated that the difference in CUL3-WT and CUL3-∆9 protein amounts could be linked to a difference in their translation rate. We subsequently normalized CUL3 quantities for all quantitative mass spectrometry experiments.

Upon induction of PrC-CUL3 or PrC-CUL3-∆9 with tetracycline, we performed label-free quantitative mass spectrometry analysis of CUL3 and CUL3-∆9 immunocomplexes (two independent experiments in Appendix A). 

### 2.2. Exon 9 Deletion Does Not Modify the Dimerization of the CUL3-Rbx1 Complex

As expected, CUL3 was the major protein identified in both purifications (Mascot scores: 6127 and 6718 (Appendix A, respectively)). The presence of tryptic peptides corresponding to exon 9 sequence in the PrC-CUL3-∆9 immunocomplexes suggested that CUL3-WT and CUL3-∆9 assembled as heterodimers. It is well established that CRL3 enzymes function as dimers [11], but whether they are CUL3-∆9 homo- or heterodimerizes has not been fully investigated. The BRET technique was used to investigate this point. Briefly, CUL3-WT and CUL3-∆9 were tagged with either a bioluminescent protein (luciferase) as donor or a fluorescent protein (YFP) as acceptor. The two tagged proteins were co-expressed with a fixed amount of the donor protein and increasing amounts of the acceptor protein. In the absence of close proximity (>10 nm), a linear curve is observed. In the case of close proximity (i.e., interaction), non-radiative energy transfer is observed between the two proteins, and the BRET signal rapidly increases to reach a plateau when all the donor protein is saturated by the acceptor protein. This specific interaction demonstrated a hyperbolic curve when the BRET signal (*Y* axis) is plotted to the [Luc]/[YFP] ratio (*X* axis). The value of [Luc]/[YFP] ratio for a 50% BRET signal (BRET_50_) could potentially reflect the affinity of the two partners [12]. We showed that the CUL3-WT homodimer presents an apparent high affinity (BRET_50_ CUL3-WT/WT = 0.24 ± 0.1 (*n* = 4)) and that CUL3-∆9 homodimerizes but also can heterodimerize with CUL3-WT, albeit with a slightly lower apparent affinity (BRET_50_ CUL3-∆9/∆9 = 1.27 ± 0.4 (*n* = 3); *p* = 0.04 and BRET_50_ CUL3WT/∆9 = 0.45 ± 0.24 (*n* = 3)) (see for details Figure 2. This observation may explain the dominant negative effect of CUL3-∆9. 

As previously shown [10], RBX1, the RING finger protein associated with CUL3, is precipitated equally by CUL3-WT and CUL3-∆9. This suggests that the ∆exon9 mutation does not interfere with RBX1 binding and that CUL3-∆9 should be able to transfer ubiquitin to substrates.

Among thousands of CUL3 potential interactors (Appendix A) and according to our selection criteria (see Materials and Methods), 43 proteins presented a significantly different interaction profile with CUL3-WT versus CUL3-∆9. Several of these proteins were further investigated for their interactions with CUL3-WT and CUL-∆9 by reciprocal co-immunoprecipitation using an epitope-tagged version of these proteins. The co-immunoprecipitation data were confirmed on the corresponding native proteins (data not shown).

### 2.3. Among Several CRL3 Regulators, Glomulin, and USP25 Interact More with CUL3-∆9

Glomulin is a generic Cullin inhibitor, which was first characterized as a protein essential for normal development of the vasculature. Mutations in this gene have been associated with glomuvenous malformations, also called glomangiomas [13]. Glomulin interferes with CRL function by masking the E2 interaction surface on the E3 moiety (RBX1) [14]. Glomulin interacted more efficiently with CUL3-∆9 with ratios 3 or 14.3-fold, according to the two mass-spectrometry experiments (Figure 3A,B). We corroborated this result by performing a reciprocal co-immunoprecipitation experiment (Figure 3C). Neddylation inhibition of PrC-CUL3-WT or ∆9 did not modify this differential interaction, suggesting a structural change in CUL3-∆9, favoring glomulin interaction.

Among CUL3-WT and ∆9 differential interactors possibly involved in contractility, we can also mention USP25, an enzyme interacting with sarcomeric proteins (actin alpha-1, filamin C, and myosin binding protein C1) [15]. USP25 is a deubiquitinating enzyme (DUB) abundantly present in both purifications, but interacting preferentially with PrC-CUL3-∆9 (4.5-fold and 3.8-fold, respectively (Figure 3A,B)). We confirmed this result by a co-immunoprecipitation experiment (Figure 3D). Indeed, USP25 strongly interacts with PrC-CUL3-∆9, suggesting that it could counteract CRL3 ubiquitinating activity. However, this strong interaction was almost suppressed by neddylation inhibition of CUL3-∆9 with MLN4924, suggesting that this interaction is dependent on the important neddylation of CUL3-∆9, which was not observed for CUL3-WT (Figure 3D). This result was confirmed by comparing the USP25 interaction to CUL3-WT or CUL3-∆9 after the mutation of the unique lysine residue, which covalently binds NEDD8 (K712R) [11]. As shown in Figure 3E, this mutation suppressed the neddylation of both CUL3-WT and CUL3-∆9 and considerably reduced the interaction of CUL3-∆9 with USP25.

Therefore, both glomulin and USP25 may participate in the loss of function of CUL-∆9 by reducing its ubiquitination capacity towards substrates. To investigate this point, we analyzed the degradation of WNK4 in the presence or absence of KLHL3 and glomulin, USP25 or USP28, a closely related DUB which does not interact with CUL3 (Appendix A). As expected, KLHL3 expression drastically increased WNK4 degradation, but more interestingly, USP25 reduced this degradation, whereas USP28 did not and glomulin had no effect.

### 2.4. COP9 Signalosome and CAND1, Two Major Regulators of CRL Cycling, Do Not Interact with CUL3-∆9

The COP9 signalosome (CSN) and CAND1 are two major CRL regulators with a central role in the dynamic assembly of CRL complexes. The COP9 signalosome (CSN) is a multiprotein complex composed of eight subunits (CSN1-CSN6, CSN7A or CSN7B and CSN8). The CSN catalyzes Nedd8 hydrolysis from the lysine residue 712 of the CUL3 WHB subdomain [4,5](Figure 1A). As expected, all CSN 1–8 subunits were recovered in CUL3 immunoprecipitates (Figure 4A). However, CSN subunits preferentially bound to PrC-CUL3-WT compared to PrC-CUL3-∆9, with ratios ranging from 2.4 to 9.4-fold for the nine CSN subunits identified by mass spectrometry (Figure 4B). We confirmed these observations using co-immunoprecipitation experiments. Whereas CSN4 readily co-immunoprecipitated CUL3, it failed to co-immunoprecipitate CUL3-∆9, and these observations were not modified when CUL3 neddylation was prevented with MLN4924, a specific inhibitor of neddylases (Figure 4C). We concluded that CUL3-∆9 was defective in the interaction with the COP9 signalosome, which was fully consistent with previous observations [10,14,15]. This observation also explained the hyperneddylation status of CUL3-∆9.

The second major regulator of the CRL3 complex is Cullin-associated and neddylation-dissociated protein 1 (CAND1). CAND1 interacts with the entire sequence of unneddylated Cullins (Figure 1A) to promote the exchange of substrate adaptors [16,17,18,19]. In our label-free mass spectrometry experiments, CAND1 bound more efficiently to PrC-CUL3-WT with ratios ranging from 10 to 49-fold as compared to CUL3-∆9 (Figure 4B).

These results were confirmed by co-immunoprecipitation experiments in the presence or absence of the neddylation inhibitor MLN4924 in order to test the role of neddylation in this interaction. As expected, CUL3-WT strongly interacted with CAND1, and this interaction was even stronger after neddylation inhibition, confirming a better interaction of the unneddylated form of CUL3. The hyperneddylated form of CUL3-∆9 did not interact with CAND1, as previously reported [10]. Surprisingly, deneddylation of CUL3-∆9 did not restore the interaction (Figure 4D) and thus suggests a structural defect of CUL3-∆9, impairing its interaction with CAND1.

Taken together, these results indicate that CUL3-∆9 is defective in the binding of the key CRL regulators CAND1 and the COP9 signalosome.

### 2.5. CUL3-∆9 Interacts More Efficiently with Numerous BTB-Domain Containing Adaptors

Among the 66 BTB-domain containing proteins identified in CUL3 immunoprecipitates (Table 1), 28 were significantly more abundant in PrC-CUL3-∆9 immunoprecipitates, with ratios ranging from 2 to 11-fold as compared to CUL3-WT (Figure 5B). The remaining 38 BTB-domain containing proteins were slightly more abundant in CUL3-∆9 (≥2-fold *n* = 9) or equally abundant between CUL3-WT and CUL3-∆9 (*n* = 24). None were more abundant in CUL3-WT immunoprecipitates. Preferential binding to some BTB adaptors did not correlate with a higher abundance of those proteins, as reflected by their Mascot scores, the absence of a correlation between these scores, and the fold differences (
Figure 5A,B).

The absence of a pool of CUL3 bound to CAND1 may explain why BTB substrate adaptors appear to be more abundantly bound to CUL3-∆9, as a result of a more abundant “free” pool of CUL3. Alternatively, the deletion of CUL3 exon 9 might increase the affinity of CUL3 to BTB proteins. We decided to investigate these two possibilities. 

Among those BTB adaptors, the three Bacurd homologues (Bacurd1, Bacurd2, and Bacurd3) are massively present in CUL3-∆9 immunocomplexes (ratios ranging from 6 to 10-fold, Table 1 and Figure 5B). Bacurd proteins act as substrate adaptors of RhoA, the degradation of which may play a crucial role in vascular contractility [20]. Using reciprocal co-immunoprecipitation, we demonstrated that PrC-CUL3-∆9 interacted significantly more with Bacurd1 than PrC-CUL3-WT, and this interaction was not significantly modified by CUL3 neddylation inhibition (Figure 5C).

Additional BRET experiments, designed to better analyze the protein interactions and their potential affinity, showed a similar interaction of Bacurd1 with CUL3-WT and CUL3-∆9, with a non-significant difference of BRET_50_ values (reflecting the affinity of the two proteins) for CUL3-WT (0.714 ± 0.16) and CUL3-∆9 (1.015 ± 0.2) (Figure 5D). This suggests that the defective ubiquitination and degradation of RhoA in cells expressing CUL3-∆9 was not due to a defective association of Bacurd to CUL3-∆9 [21].

This study was extended to two other BTB-domain proteins, i.e., KLHL21, which regulates cell cycle and cell motility [22,23], and KLHL7, the mutations of which are involved in retinitis pigmentosa [24,25]. The increased interaction of KLHL21 with CUL3-∆9 that was observed in mass spectrometry experiments (6.3-fold, *p* = 0.03 and 3.7-fold *p* = 0.01 in the two experiments (Table 1)) was confirmed by co-immunoprecipitation experiments (Figure 6A). 

KLHL7 was not differentially associated with CUL3-WT and CUL3-∆9 as assessed by the two label-free interactome studies (1.31-fold, *p* = 0.07 and 1.34-fold *p* = 0.17) (Table 1). BRET experiments identified no significant difference in the apparent affinity of KLHL7 for native CUL3-WT and CUL3-∆9 (BRET_50_= 1.068 ± 0.37 and 2.708 ± 0.89, respectively; *p* = 0.195, *n* = 4) as for unneddylated forms of CUL3-WT and CUL3-∆9 (BRET_50_= 1.272 ± 0.45 and 2.408 ± 0.66, respectively; *p* = 0.226, *n* = 3) (Figure 6B). We also identified other notable BTB proteins interacting preferentially with CUL3-∆9. The list included KCTD20 and its paralogue BTBD10, which are positive Akt regulators, and ENC1, involved in ciliogenesis in zebrafish [23,26,27,28,29]. 

A similar analysis was conducted for the BTB-domain protein, KLHL3, which is expressed exclusively in the distal convoluted tubules of the kidney (and therefore not present in our mass spectrometry experiments). KLHL3 is the essential adaptor of CUL3 for WNK ubiquitination and degradation and therefore FHHt pathology. As shown in Figure 7, both CUL3-WT and CUL3-∆9 interacted with KLHL3 in co-immunoprecipitation (Figure 7A) and BRET experiments (Figure 7B). The BRET_50_ was significantly lower for CUL3-∆9 (0.9115 ± 0.33) than for CUL3-WT (1.855 ± 0.2) (*p* = 0.0478, *n* = 3). This two-fold difference in the BRET_50_ was abolished by pretreating cells with MLN4924 (BRET_50_ = 0.9143 and 1.172 for CUL3-WT and CUL3-∆9, respectively) (Figure 7C), suggesting a better affinity of the neddylated forms of CUL3 for KLHL3, but no difference in this affinity between CUL3-WT and CUL3-∆9. As seen for CUL3 (Figure 2) and Bacurd (21), the expression of the mutant form of CUL3 did not modify the degradation of KLHL3 (Figure 7D). In addition, the expression of KLHL3 with CUL3-∆9 reduced the degradation of WNK4 compared to CUL3-WT (Appendix A), as demonstrated previously by several authors [9,30,31].

### 2.6. Exploring the Dynamic Nature of CUL3-∆9-Associated Proteome by SILAC-Based Quantitative Mass Spectrometry

CUL3-∆9 is defective in the binding to CAND1, the protein that allows adaptor exchange cycles on Cullins [17,18,19]. CUL3-∆9 is thus predicted to be defective in the exchange of BTB adaptor subunits. Consistent with this hypothesis, our quantitative label-free proteomic analysis indicated that several BTB proteins were preferentially associated with CUL3-∆9 in cellulo (Figure 5B and Figure 6, and Table 1), and this preferential association was not merely due to an increase in the affinity of BTB proteins for CUL3-∆9 (Figure 5D, Figure 6B and Figure 7B). We thus hypothesized that CUL3-∆9 might be a trap for some BTB adaptors. To test this possibility, we needed a method that allowed measuring the steady state and dynamic nature of CUL3-WT and CUL3-∆9 associated proteomes in order to monitor the dynamic exchange of BTB proteins with CUL3-WT and CUL3-∆9, respectively. Stable Isotope Labeling by Amino-acids in Cell culture (SILAC) was used, and a dynamic pulse-chase experiment followed by mass spectrometry analysis of CUL-3 and CUL3-∆9 immunocomplexes was performed (Figure 8A). A similar type of experiment has previously been used to explore the F-box protein exchange activity of CAND1 [17,18].

Briefly, we first transiently induced 293 cell lines with tetracycline in order to produce PrC-tagged CUL3-WT or CUL3-∆9. These cells grew in a normal “light” medium for 24 h, before being switched to a medium formulated with isotopically heavy lysine and arginine (“heavy” medium). After 6 h in “heavy” medium, cells were lysed, and then CUL3-WT- and CUL3-∆9-associated proteomes were analyzed by quantitative mass spectrometry. An analysis of isotopic ratios revealed that the CUL3-∆9 mutant was compromised in adaptor exchange since the newly synthesized “heavy” BTB-adaptors were less abundant in CUL3-∆9 immunocomplexes as compared to CUL3-WT (13/21 BTB-adaptors, Figure 8B). This effect could be under-estimated because of the existence of CUL3-WT/CUL3-∆9 dimers, the CUL3-WT moiety being responsible for the exchange of only one adaptor in the heterodimer. Those results imply that CUL3-∆9 could inhibit CUL3-WT normal function by sequestering some BTB adaptors, thus disturbing their “free” pool inside the cell.

## 3. Discussion

Since the discovery of CUL3 mutations responsible for FHHt pathophysiology [1], several groups have tried to elucidate the mechanism by which the deletion of exon 9 (amino-acids 403 to 459), the common consequence of these mutations, results in a loss of function of the CRL3 complex, leading to decreased ubiquitination of substrates such as the WNKs or RhoA. This loss of function is contradictory to the concordant observation, confirmed in the present study, that the CUL3-∆9 mutant presents an increased neddylation, a post-translational modification necessary for cullin activation. The proposed hypotheses to reconciliate these observations include 1) the accelerated degradation of substrate adaptors (i.e., BTB proteins), which are necessary for substrate ubiquitination by the cullin, 2) increased flexibility of the CUL3-∆9 molecule, or 3) modulation in the interaction of several cullin regulators [8,9,10,30,31,32,33,34]. However, data supporting those hypotheses are often contradictory and sometimes purely theoretical.

Cullin-Ring Ligases (CRL) are multi-subunit E3 ubiquitin ligases nucleated around a cullin scaffold protein controlling the ubiquitin-mediated proteasomal degradation of a large number of protein substrates. Cullin 3 (CUL3), as all cullin proteins, is composed of three cullin-repeats (CR1-CR3) followed by a four-helix-bundle (4HB) subdomain corresponding to the exon 9 sequence at its N-terminus (Figure 1A). The first cullin-repeat (CR1) specifically interacts with a large variety of substrate adaptors containing a BTB/POZ domain (for Bric-a-brac, Tramtrack, and Broad Complex/Pox virus and Zinc finger, hereafter referred to simply as BTB) [3] and an associated protein–protein interaction domain allowing versatile substrate binding. For instance, the KLHL3 adaptor interacts with CUL3 via its BTB domain and recruits its substrates via Kelch repeats. The C-terminal part of CUL3 comprises a cullin/RING (CR) subdomain that intertwines sequences of cullin and the RING finger protein RBX1, which mediates ubiquitin transfer from the E2 enzyme to the substrate. Finally, this part of CUL3 contains a winged-helix B (WHB) subdomain (Figure 1A) interacting with neddylases, which catalyze the covalent binding of NEDD8 to Lys 712 and are pharmacologically inhibited by MLN4924. 

Here, we have provided the most extensive proteomic study to date of WT and mutant CRL3 complexes performed in human cells. The systematic nature of this analysis is highlighted by the abundance of BTB adaptors and CRL regulators identified in our screen compared to previous similar approaches [35,36]. In addition, this study provided a quantitative comparison between the interactomes of CUL3-WT and CUL3-∆9, including the classical N-terminal binding BTB substrate adaptors and the C-terminal binding RBX RING protein, which binds the E2 ligases and the regulatory proteins, including CAND1, COP9 signalosome, and others.

BTB adaptors, which classically bind to both the N-terminus of CUL3 and to the substrates, were analyzed for differential interaction with CUL3-WT and CUL3-∆9. Among the 66 BTB proteins identified in the label-free differential interactomes, all bound at least equally to CUL3-WT and CUL3-∆9 and half of them bound with a strong preference to CUL3-∆9. According to BRET experiments performed for BACURD, KLHL7, and KLHL3, this difference of interaction appeared not to be dependent on a difference in affinity. Finally, and interestingly, preferential binding to some BTB adaptors did not correlate with a higher abundance of those proteins, as reflected by their Mascot scores and the absence of correlation between these scores and the fold differences. This capacity of CUL3-∆9 to strongly bind BTB adaptors was first shown for KLHL3, the BTB adaptor present in the distal convoluted tubule of the nephron, where it is responsible for WNK degradation [9]. Other BTB-domain proteins have been identified as interacting more strongly with CUL3-∆9 including Bacurd1 and RhoBTB1, two BTB adaptors possibly involved in RhoA stability [8], and NUDCD3, a KELCH-domain directed co-chaperone for HSP90 [37]. Up to now there is no clear explanation for the stronger interaction of numerous BTB domain proteins to CUL3-∆9 as compared to CUL3-WT. One possible but partial explanation is the increased abundance of “free” CUL3-∆9, consecutively to its absence of binding to CAND1. 

It has been shown that KLHL3 is less abundant in cells transfected with CUL3-∆9 compared to CUL3-WT [9,10,33] and that both CUL3-∆9 and KLHL3 are ubiquitinated when CUL3-∆9 is expressed, which is not the case when CUL3-WT is expressed [10]. These observations were confirmed by cycloheximide chase assay and Western blot showing that KLHL3 is a very stable protein (half-life > 24 h in the presence of CUL3-WT) but presents a shorter half-life in the presence of CUL3-∆9 [30]. These data lead to the hypothesis that a “hyperactive” CUL3-∆9 may ubiquitinate substrate adaptors instead of substrates, thus explaining the defect in substrate degradation. In the present study, using a cycloheximide chase assay on bioluminescent-tagged KLHL3, CUL3-WT, or CUL3-∆9 proteins, we did not observe any difference in the half-lives of CUL3-WT, CUL3-∆9, or KLHL3 in the presence of either WT or mutated CUL3. The degrading effect of CUL3-∆9 on substrate adaptors was not observed for Rho-BTB and Keap1 [30] or for KLHL2, KLHL16 and Keap1 [34]. Finally, CUL3-∆9-mutated mice confirmed that there was no change in KLHL3 amounts in the kidney when CUL3-∆9 was expressed [10]. Altogether, these data raise some doubts about the physiological role of BTB-domain protein degradation in the loss-of-function mechanism of CUL3-∆9.

COP9 signalosome and CAND1 are two important elements of the regulation cycle of CRL complexes [17]. The CAND1/CUL3 complex is inactive, and its dissociation is necessary for the binding of substrate adaptor and substrate to CUL3. The association of CUL3 to substrate adaptor and substrate is required for the neddylation of CUL3 by neddylases such as E1 APPBP1 Uba3 and Ubc12 enzymes [38]. This neddylation is a necessary (but maybe not sufficient) step for the activation of the CRL3 complex, which is now able to ubiquitinate its substrate. Inactivation of the CRL3 complex is initiated by the CUL3 deneddylation by the COP9 signalosome, followed by dissociation of the substrate adaptor and reassociation to CAND1 (Figure 9).

The interaction defect of CUL3-∆9 with the COP9 signalosome as assessed by the absence of interaction of CUL3-∆9 with several of its subunits, including CSN4 (present study), CSN5/Jab1 [30,39], and CSN8 [10], explains the hyperneddylation state of CUL3-∆9. As an attempt to identify the CUL3 domain involved in CSN interaction, Cornelius et al. produced several GST-fused domains of CUL3 and analyzed their interaction with CSN5. The domain including amino-acids 461–586 is the binding domain to CSN5 and is adjacent to the exon 9 domain (amino-acids 403–459), indicating that the absence of binding to CSN of CUL3-∆9 is more the consequence of a secondary folding change than a direct deletion of the binding domain to CSN [30].

The absence of interaction between CUL3-∆9 and CAND1 also interrupts the regulation cycle of CUL3. CAND1 binds to both the N-terminal and C-terminal parts of CUL3 [16]. At the N-terminus, it binds to the substrate adaptor binding site, explaining the competing effect of CAND1 for the substrate adaptor binding. At the C-terminus, it binds to a surface of CUL3, which contains the potentially neddylated lysine, explaining why the neddylated form of CUL3 cannot bind to CAND1. However, it remains unclear whether the 403–459 amino-acid sequence participates directly at this binding site. 

Briefly, CUL3 is involved in a dynamic cycling of assembly/disassembly with adaptor, substrate, and CAND1 necessary for the rapid and adaptative exchange of substrates to be ubiquitinated, maybe favoring the binding of adaptors for which the substrate is available, as previously suggested [17]. The CUL3-∆9 mutant is defective in this dynamic cycling and therefore presents as a “non-generic” BTB adaptor trap that likely affects the intracellular “free pool” of some specific adaptors (Figure 9). Indeed, our SILAC screen highlighted for the first time the dynamic nature of BTB adaptor cycling in CRL3 complexes, as well as the inhibition of BTB adaptor exchange in the context of the CUL3-∆9 mutant. This inhibition, together with the existence of CUL3-WT-CUL3-∆9 heterodimers, provides a comprehensive model for the dominant negative nature of this mutant, rather than a haploinsufficiency hypothesis.

In summary, the Cullin 3-∆9 mutant protein interacts with its classical partners (BTB-proteins, RBX) in the CRL3 complex but is mostly inactive, as assessed by its inability to ubiquitinate its physiological substrates, i.e., WNK, RhoA, and others. Four hypotheses emerged in the past few years to explain this loss of function: (i) Haploinsufficiency of CUL3 was suggested by the Kurtz group because CUL3-∆9 triggers its own degradation [10]. However, several mouse models of heterozygous KO of CUL3 do not recapitulate the FHHt phenotype [31,40], excluding this hypothesis. In addition, our data are not in favor of this hypothesis. It is now clear that the CUL3-∆9 mutant has a dominant-negative effect. (ii) The dominant-negative effect may be the consequence of an accelerated degradation of CUL3 adaptors, as suggested by several groups [10]. However, increased degradation was not observed in vitro or in vivo, neither for KLHL3 nor for other adaptors [10] nor in our study. (iii) A third hypothesis suggests an increased flexibility of CUL3-∆9, which impairs its ubiquitination capacity toward substrates. This hypothesis, based on 3D-modeling of CUL3 with the CUL1 crystal structure as a model [10], was not demonstrated experimentally. (iv) Finally, our interactome data suggest a new hypothesis involving the dynamic regulation CRL complexes. CUL3-∆9 is deficient in binding CAND1 and the COP9 signalosome, which impairs dynamic cycling of the complex and rapid exchange of adaptors and substrates, as highlighted by our SILAC experiments. CUL3-∆9 would thus behave as an inactive BTB-adaptor trap. 

In conclusion, our exhaustive differential proteomic screen of CUL3-WT and CUL3-∆9 highlights different facets of CUL3 regulation, which might vary in different tissues and cell types, thus opening exciting avenues of investigation for further in vitro and in vivo studies in FHHt.

## 4. Materials and Methods

### 4.1. Vectors

pcDNA-hCUL3 encoding HA-tagged human CUL3 is a gift from C. Rochette-Egly (Department of Functional Genomics and Cancer, University of Strasbourg, Illkirch, France). Deletion of exon 9 and the K712R mutation of CUL3 were generated using the QuikChange Lightning Site-Directed Mutagenesis Kit (Agilent Technologies, Les Ullis, France) and checked by direct sequencing. Flag-hKLHL3 cDNA was purchased from the MRC-PPU facility of the University of Dundee (UK). WNK4 constructions were described previously [41]. pFLAG-CSN4 was provided by LP [42]; pGEX-TEVsite-human glomulin was a gift from Brenda Schulman (Addgene plasmid # 52292, Watertown, MA, USA). GFP-USP25 was a gift of Gemma Marfany (University of Barcelona, Barcelona, Spain) and USP28 was a gift of François Leteurtre, CEA, Paris-Saclay, France).

CUL3-WT and CUL3-∆9 and hKLHL3 cDNA were subcloned into pEYFP, phLuc, and pNanoLuc vectors and into pCDNA5/FRT/(His)6-Protein C vector (derived from pCDNA5/FRT/V5-His (Invitrogen, Paris, France) as described [43]. Glomulin cDNA (from vector pGEX-TEVsite-human Glomulin) was also subcloned into pCDNA5/FRT/(His)6-Protein C vector. All tags are fused at the N-terminus of cDNAs.

### 4.2. Cell Culture and Transfection

Flp-In^TM^ T-REx^TM^ 293 cells (Invitrogen, Paris, France) were stably transfected with CUL3-WT, CUL3-∆9, and hGlomulin (GLMN) or empty pCDNA5/FRT/(His)6-Protein C vectors following the manufacturer’s instructions. The stable and inducible cell lines were grown in DMEM medium (Gibco, Paris, France) supplemented with 10% (*v*/*v*) Fetal Calf Serum (FCS) (Fisher Scientific, Ilkirch, Frnace), Penicillin/Streptomycin (0.1 mg/mL each) (Gibco, Paris, France), Hygromycin B 200 μg/mL (Invivogen, Toulouse, France), and Blasticidin 7.5 μg/mL (Invivogen Toulouse, France). (His)6-Protein C-CUL3, CUL3-∆9, Flag-KLHL3 and GLMN were induced with 10 μg/mL tetracycline (Sigma; Saint Quentin Falavier, France). For transient expression of myc-CAND1, FLAG-CSN4, GFP-USP25, GFP-Bacurd, and Flag-KLHL3 cells were transfected using Effectene^®^ (Qiagen, Marseille, France) or Jetoptimus+ (Polyplus-transfection, Illkirch-Graffenstaden, France) following the manufacturer’s instructions. 

### 4.3. Interactome Analysis Using LC-MS/MS

#### 4.3.1. Samples Preparation

Flp-In^TM^ T-Rex^TM^ 293 cells (Invitrogen, Paris, France), stably transfected with CUL3-WT, CUL3-∆9, or empty vector are induced by tetracycline (10 µg/mL) for 16 h. Then, cells were trypsinized, washed in PBS and lysed with 50 mM HEPES pH 7.5, 150 mM NaCl, 0.5% Nonidet P-40, 2 mM CaCl2, 5% glycerol, 2 mM 1.10-ortophenantroline, supplemented with an EDTA-free protease inhibitor cocktail (Roche Diagnostics, Meylan, France). Lysates were incubated overnight at 4 °C with HPC4-sepharose (Roche, Meylan, France) beads, pre-equilibrated with lysis buffer. After three washes in washing buffer (50 mM HEPES pH 7.5, 150 mM NaCl, and 1 mM CaCl_2_, supplemented with a protease inhibitor cocktail (Roche Diagnostics, Meylan, France)), samples were analyzed. In each of the 2 independent experiments, the 3 samples (CUL3-WT, CUL3-∆9, and empty vector) were compared.

For pulse chase experiments with heavy amino-acids, the same cell lines were induced with tetracycline in a normal “light” medium for 24 h, before switching them to a medium formulated with isotopically heavy lysine and arginine (“heavy” medium). After 6h of incubation in “heavy” medium cells were lysed, Pr-C CUL3 was immuno-precipitated and the immunocomplexes were analyzed by SILAC quantitative mass spectrometry. Two independent biological experiments were analyzed in the same SILAC experiments, and results are expressed for each protein as the mean of the 2 biological replicates.

#### 4.3.2. LC-MS/MS Acquisition 

Samples were digested with trypsin (0.2 μg/μL) in NH4HCO3 25mM buffer with (or without for experiment 1) 10% acetonitrile overnight at 37 °C. Peptides were desalted using ZipTip C18 Pipette Tips (Thermo Fisher Scientific, Les Ullis, France) and analyzed by an Orbitrap Q-exactive Plus mass spectrometer (or a LTQ Orbitrap Velos mass spectrometer for experiment 1) in positive mode (Thermo Fisher Scientific, Les Ullis, France) coupled to a Nano-LC Proxeon 1000 equipped with an EASY-spray ion source (Thermo Fisher Scientific, Les Ullis, France). Peptides were separated by liquid chromatography with the following parameters: Acclaim PepMap100 C18 pre-column reversed phase (2 cm, 3 μm, 100 Å) (5 cm, 300 μm i.d., 100 Å for experiment 1), EASY-spray C18 column reversed phase (P/N ES803A, 50 cm, 75 μm i.d., 2 μm, 100 Å), 300 nL/min flow rate, gradient from 95% solvent A (water, 0.1% formic acid) to 35% (40% for experiment 1) solvent B (100% acetonitrile, 0.1% formic acid) over a period of 98 min (100 min for experiment 1), followed by a column regeneration of 20 min, giving a total run time of 118 min (120 min for experiment 1). Peptides were analyzed in the Orbitrap cell, in full ion scan mode, at a resolution of 70,000 (30,000 for experiment 1) with a mass range of *m/z* 375–1500 (400–1800 for experiment 1) and an AGC target of 3 × 10^6^. Fragments were obtained by Higher-energy C-trap Dissociation (HCD) (Collisional-Induced Dissociation (CID) for experiment 1) activation with a collisional energy of 27% (40% for experiment 1) and a quadrupole isolation window of 1.4 *m/z* (1 Da for experiment1). MS/MS data were acquired in the Orbitrap with a resolution of 17,500, a TopN of 20, with an AGC target of 2 × 10^5^ and with a dynamic exclusion of 30 s (in the Ion trap, a TopN of 20 with a dynamic exclusion of 20 s for experiment 1). Peptides with charge states of 2 to 4 or more were included for the acquisition (charge states = 2 to 8 for experiment 1). The maximum ion accumulation times were set to 50 ms (100 ms for experiment 1) for MS acquisition and 45 ms (50 ms for experiment 1) for MS/MS acquisition. 

#### 4.3.3. LC-MS/MS Data Processing and Analysis

Label-free quantification was performed on Progenesis QI for Proteomics (Waters, Milford, MA, USA) in Hi-3 mode for protein abundance calculation. Proteins were filtered with a fold change ≥ 2 and a *p*-value ≤ 0.05. MGF peak files from Progenesis were processed by Proteome Discoverer 1.4 with the Mascot search engine (Version 2.5.1). The Swissprot database (release 2014_06, 545,657 entries) with *Homo sapiens* taxonomy (20214 entries) was used. A maximum of 2 missed cleavages was authorized. Precursor and fragment mass tolerances were set, respectively, to 7 ppm and 0.5 Da for the Orbitrap Fusion, and 6 ppm and 0.02 Da for the Orbitrap Qexactive Plus. The following post-translational modifications were included as variables: Acetyl (Protein N-term, K), Oxidation (M), Phosphorylation (STY), LRGG(K), EQIGG (K), and Deamidation (NQ). Spectra were filtered using a 1% FDR using the percolator node. No fixed modification was considered.

Pulsed SILAC quantification was performed on Proteome Discoverer 1.4 using the Mascot search engine with the same parameters as the Label-free quantification. Quantification was done only with unique peptides with a fold-change ≥ 2. Precursor and fragment mass tolerances were set, respectively, to 6 ppm and 0.02 Da. The following post-translational modifications were included as variables: Acetyl (Protein N-term, K), Oxidation (M), Phosphorylation (STY), Label 13C6 (R), Label 13C6 15N2 (K), and Deamidation (NQ). Spectra were filtered using a 1% FDR using the percolator node.

### 4.4. Reciprocal Co-Immunoprecipitations of CUL3-WT and CUL3-∆9

#### 4.4.1. Immunoprecipitation

Cells were induced (or not) with tetracycline (10 µg/mL) 32 h post-transfection, treated (or not) with MLN4924 (1 µM) (Millenium Pharmaceuticals, Cambridge, MA, USA) 42 h post-transfection, harvested 48 h post-transfection, washed in cold PBS, and frozen in liquid nitrogen. Cell pellets were lysed for 1 h at 4 °C in IP lysis buffer (50 mM HEPES pH 7.5, 150 mM NaCl, 0.5% Nonidet P-40, 2mM CaCl2, 5% glycerol, 2 mM 1.10-ortophenantroline, supplemented with a protease inhibitor cocktail (Roche Diagnostics, )) and centrifuged at 15,000× *g* for 30 mn at 4 °C. Cell lysates (supernatants) were incubated overnight at 4 °C with HPC4-sepharose (Roche, Meylan France), GFP Trap-agarose (Chrometek, Planneg, Germany), or the appropriate antibody at a 1/500 dilution (antiFLAG (Sigma, Saint Quentin Falavier, Lyon, France), anti-myc (Cell signaling, Leiden, Holland)) followed by Protein G or A mag Sepharose (Healthcare GE, Velizy, France) for 1 h. All types of beads are pre-equilibrated with lysis buffer. After three washes in washing buffer (50 mM HEPES pH 7.5, 150 mM NaCl and 1 mM CaCl_2_, supplemented with a protease inhibitor cocktail (Roche Diagnostics, Meylan France)), bound proteins were separated by SDS–PAGE.

#### 4.4.2. Immunoblotting

Lysates and immunoprecipitates were analyzed by SDS-PAGE on mini-PROTEAN Stain-free Precast gels (Biorad, Marne la coquette, France), transferred to nitrocellulose membrane, and immunoblotted with primary antibodies, including anti-FLAG M2 (Sigma, Saint Quentin Falavier, France), anti-myc (9B11, Cell Signaling, Leiden, Holland), polyclonal anti-protein C (hpc4, Cell signaling), rabbit anti-GAPDH (Abcam, Cambridge, UK), polyclonal anti-GFP (gift from S. Miserey-Lenkei, Curie Institute; Paris, France), and polyclonal anti-KLHL21 (Invitrogen, Paris, France) antibodies. Thereafter, the membranes were incubated with a horseradish peroxidase-conjugated mouse or rabbit secondary antibody (1:5000 dilution). The images were obtained with chemiluminescence (Clarity Max Western EXL, Biorad; Marne la coquette, France) using a luminescent image analyzer (ChemiDoc XRS+, Biorad, Marne la coquette, France) and quantified with ImageLab software version 8.1.0 (Biorad, Marne la coquette, France).

### 4.5. Degradation and BRET Experiments

For degradation assays, cells were transfected with nanoluc or luciferase tagged CUL3, CUL3-∆9, KLHL3, or Flag-WNK4 constructs and 24 h after transfection, cells were treated with cycloheximide (20 μg/mL) (Sigma; Saint Quentin Falavier, France). Cells were collected at different time points after cycloheximide addition and then lysed in passive lysis buffer (Promega, Charbonnières les Bains, France), and luminescence was measured in presence of coelenterazine H or NanoGlo (Promega, Charbonnières les Bains, France) or lysed in IP lysis buffer and analyzed by Western blot as described above.

The apparent affinity of WT and ∆9 CUL3 for themselves or Bacurd, KLHL7, and KLHL3 was evaluated by BRET. In each experiment, a fixed amount of BRET donor plasmid (CUL3-, Bacurd-, KLHL7-luciferase) was transfected in HEK293 cells (6-well plates) in association with increasing amounts of the BRET acceptor (YFP-CUL3-WT or −∆9). Signals were measured using a Mithras LB 940 multimode reader (Berthold, Thoiry, France). BRET results were expressed in milli-BRET units (mBRET), or % of maximal BRET signal plotted as a function of YFP/Rluc ratio, in which YFP represents the actual amount of expressed BRET acceptor and Rluc the amount of BRET donor in each sample.

### 4.6. Statistical Analyses

For label-free mass spectrometry experiments, the reliability of the quantification measurements was handled with an Anova test for each quantified protein using the Progenesis QI for Proteomics software (Waters, Milford, MA, USA). For SILAC experiments, the ratios of the heavy/light measured amount of each peptide were calculated and the mean of the 2 biological experiments was performed for each protein; then, the *p*-values of peptides were calculated using the percolator algorithm, and a 1% filter was applied as a false-discovery rate threshold.

For co-immunoprecipitation experiments, statistical comparisons between 2 conditions were made using the Krustal–Wallis test (quantitative scale). Other statistical analyses used the Student’s *t* test.

## Figures and Tables

**Figure 1 ijms-23-05151-f001:**
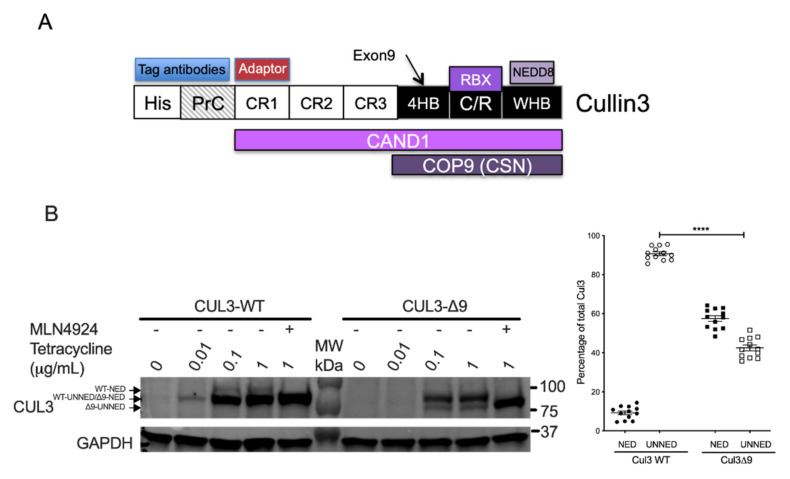
General structure of CUL3 and production of cell lines expressing CUL3-WT and CUL3-∆9. (**A**). CUL3 protein comprises several domains including (from the N- to the C-terminus): three tandem cullin-repeat domains (CR1-CR3), a four-helix-bundle domain (4HR), which includes exon 9-coded sequence, followed by a cullin/ring intermolecular domain (C/R), which intertwines the cullin 3 and RBX proteins, and finally the winged-helix B (WHB) C-terminal domain, which contains the neddylated lysine 712. For the expression in 293 cells, two short tags (6xHis and protein C tags) were added at the N-terminus in order to be recognized by specific antibodies. The different proteins interacting with CUL3 are indicated as aligned with their corresponding CUL3-interacting domain(s). (**B**). The cDNA sequence of CUL3-WT or CUL3-∆9 was inserted in the pcDNA5/FRT/TO/(His)6-Protein C vector and therefore tagged at its N-terminus with the Protein C peptide (PrC) and the 6xHis tags. These constructs were stably transfected in Flp-In^TM^ T-REx^TM^ 293 cells (Invitrogen). The expression of PrC-CUL3-WT and PrC-CUL3-∆9, induced by 0.01 to 1 μg/mL tetracycline or not, was assessed by Western blot using the polyclonal HPC4 antibody (Cell signaling) as primary antibody. Neddylated (NED) and unneddylated (UNNED) forms of CUL3-WT and CUL3-∆9 were quantified on several blots (*n* = 13) and the ratio of each form was plotted as a percentage of total expression. The percentage of neddylated (11.6 ± 2.5%) versus unneddylated (88.4 ± 2.5%) form of CUL3-WT was significantly different (**** = *p* < 0.0001) from that of CUL3-∆9 (59 ± 2% and 41 ± 2%, respectively) (*t*-test).

**Figure 2 ijms-23-05151-f002:**
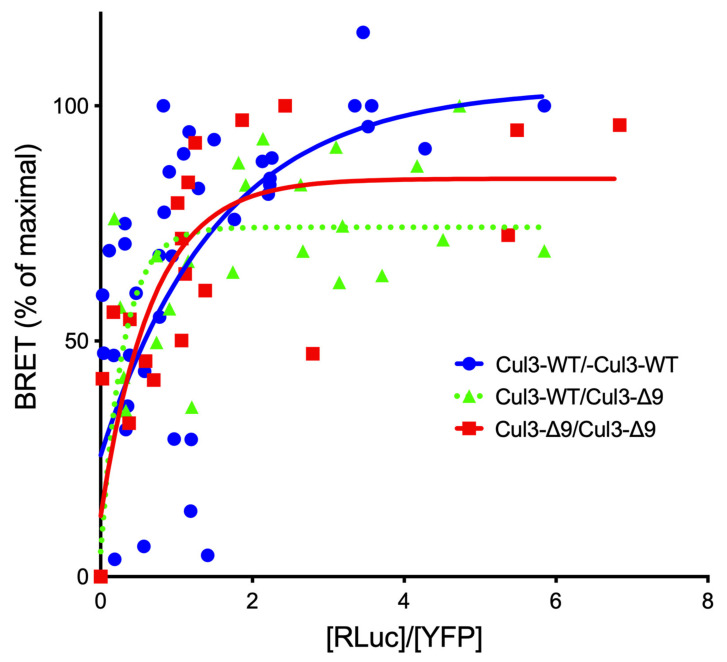
Dimerization of CUL3-WT and CUL3-∆9. CUL3-WT and CUL3-∆9 were tagged with luciferase or YFP at their C- or N-terminus and expressed in HEK293 cells. BRET signals between the donor constructs (CUL3-Luc and Luc-CUL3-∆9) and increasing amounts of the acceptor constructs (YFP-CUL3 and YFP-CUL3-∆9) were measured using a Mithras LB 940 multimode reader (Berthold). The results are expressed as percentages of the maximal BRET signal (*Y*) as a function of the Luc/YFP concentration ratio (*X*). Similar hyperbolic curves were observed for CUL3-WT and CUL3-∆9 homodimerizations and CUL3-WT/∆9 heterodimerization. BRET_50_ values, which reflect the affinity between the two interactors, were 0.24 ± 0.1 (*n* = 4), 1.27 ± 0.42 (*n* = 3), and 0.45 ± 0.24 (*n* = 3) for WT/WT, ∆9/∆9, and WT/∆9, respectively (non-significant differences except for WT/WT versus ∆9/∆9 homodimerizations (*p* = 0.041) (Student’s *t* test.)).

**Figure 3 ijms-23-05151-f003:**
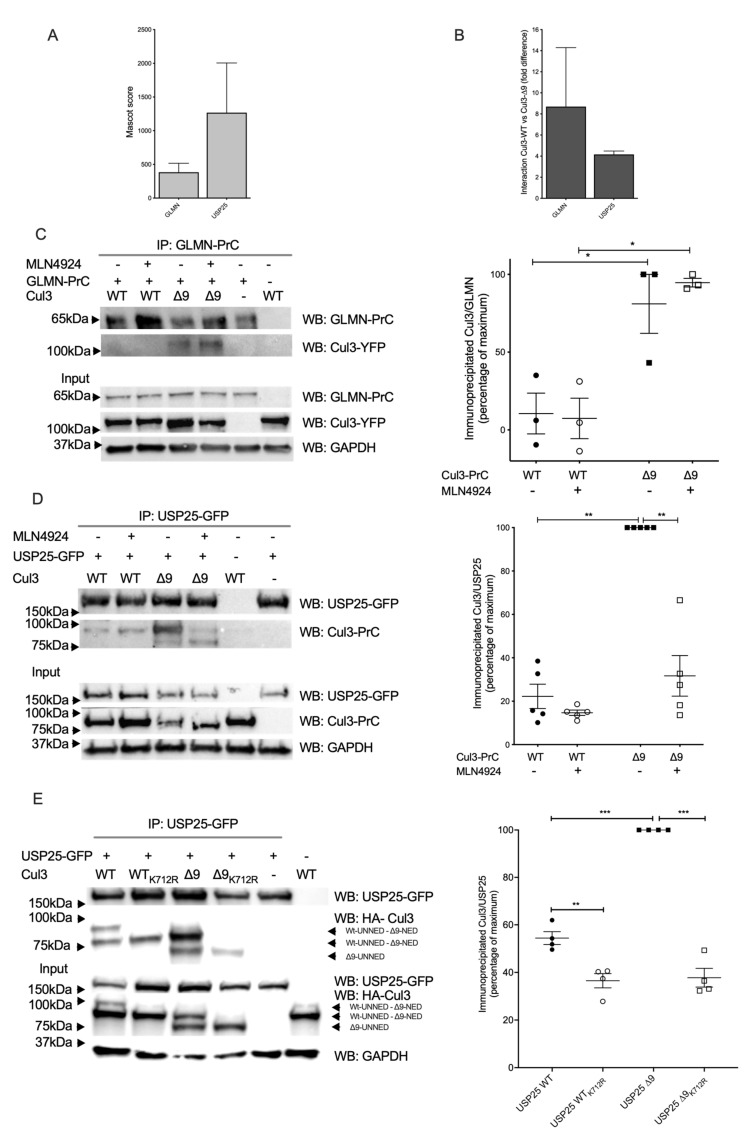
Glomulin and USP25 interact more with CUL3-∆9. (**A**). and (**B**). PrC-CUL3-WT and PrC-CUL3-∆9 were immunoprecipitated with HPC4 antibody, and the immunoprecipitate was analyzed by mass spectrometry. The abundance, as assessed by the Mascot score (**A**), and the difference in the interaction of glomulin (GLMN) and USP25 (**B**) with PrC-CUL3-WT versus PrC-CUL3-∆9 are shown. (**C**). CUL3-WT-YFP and CUL3-∆9-YFP constructs were expressed in Flp-In^TM^ T-REx^TM^ 293 cells (Invitrogen) stably transfected with GLMN-PrC. Cells were induced (or not) with tetracycline (10 µg/mL) 32 h post-transfection, and treated (or not) with MLN4924 (1 µM) (Millenium Pharmaceuticals) 42 h post-transfection. Co-immunoprecipitation of GLMN-PrC was performed using HPC4-agarose beads (Roche), followed by SDS-PAGE and transfer to nitrocellulose. On the left is shown a representative gel of the co-immunoprecipitation experiment blotted with the HPC4 (GLMN) and GFP (CUL3-WT or CUL3-∆9) antibodies. On the bottom is shown a Western blot of the same cellular extracts for CUL3, GLMN and GAPDH. On the left is presented a scatter plot of the results obtained for three experiments. The results are expressed as the percentage of the maximal ratio of immunoprecipitated CUL3/GLMN normalized on the CUL3/GAPDH ratio observed on the input gel. This ratio is significantly higher for CUL3-∆9 compared to CUL3-WT (* = *p* < 0.05) but independent of the neddylation state. (**D**). USP25-GFP construct was transfected in cells stably expressing PrC-CUL3-WT or PrC-CUL3-∆9. Cells were induced (or not) with tetracycline (10 µg/mL) 32 h post-transfection, and treated or not with MLN4924 (1µM) (Millenium Pharmaceuticals) 42 h post-transfection. Co-immunoprecipitation of USP25-GFP was performed by GFP-Trap, followed by SDS-PAGE and transfer to nitrocellulose. On the left, a representative gel of the co-immunoprecipitation experiment blotted with the HPC4 (CUL3) and GFP (USP25) antibodies is shown. On the bottom, a Western blot of the same cellular extracts for CUL3, USP25, and GAPDH is shown. On the left, a scatter plot of the results obtained for five experiments is presented. The results are expressed as the percentage of the maximal ratio of immunoprecipitated CUL3/USP25 normalized on the CUL3/GAPDH ratio observed on the input gel. This ratio is significantly higher for CUL3-∆9 compared to CUL3-WT (* *p* < 0.05; ** = *p* < 0.01), but deneddylation of CUL3-∆9 significantly reduced this interaction (* = *p* < 0.05). Statistical comparisons used the Kruskal–Wallis test. (**E**). USP25-GFP with or without HA-CUL3-WT or mutants (∆9, K712R or ∆9 + K712R) were transfected in HEK293 cells. Co-immunoprecipitations and Western blots were performed, and the results are expressed as in D. *** = *p* < 0.001

**Figure 4 ijms-23-05151-f004:**
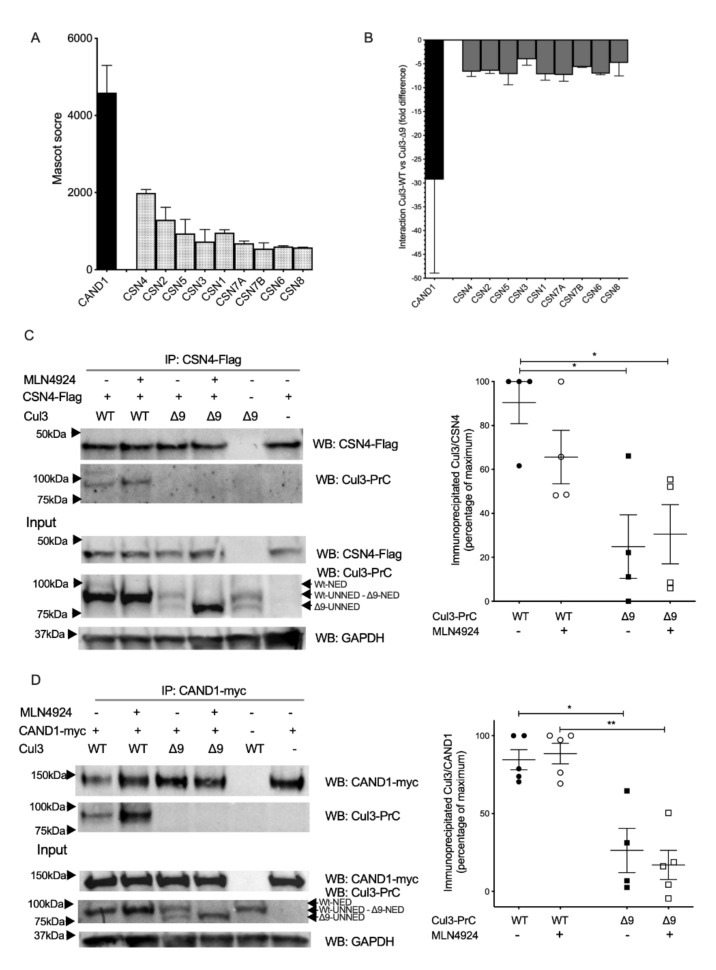
CUL3-WT and CUL3-∆9 interactions with the COP9 signalosome and CAND1. PrC-CUL3-WT and PrC-CUL3-∆9 were immunoprecipitated with HPC4 antibody, and the immunoprecipitated was analyzed by mass spectrometry. The abundance, as assessed by the Mascot score (**A**), and the difference in the interaction of each protein of the COP9 signalosome and CAND1 (**B**) with CUL3-WT and CUL3-∆9 are shown. CSN4-Flag (**C**) and CAND1-myc (**D**) constructs were transfected in PrC-CUL3-WT or PrC-CUL3-∆9-inducible cell lines (see Materials and methods). Cells were induced (or not) with tetracycline (10 µg/mL) 32 h post-transfection and treated (or not) with MLN4924 (1 µM) (Millenium Pharmaceuticals) 42 h post-transfection. Co-immunoprecipitation of CSN4-Flag or CAND1-myc was performed by incubating O/N the cell extracts with the corresponding Flag (M2, Sigma) and myc (9B11, cell signaling) monoclonal antibodies and protein G mag Sepharose beads (Healthcare GE) for 1 h, followed by SDS-PAGE and transfer to nitrocellulose. The left part shows a representative gel of the co-immunoprecipitation experiment, blotted with the HPC4 (CUL3) and Flag (CSN4) antibodies for C and HPC4 (CUL3) and myc (CAND1) antibodies for D. The bottom shows a Western blot of the same cellular extracts for CUL3, CSN4 (C), or CAND1 (D) and GAPDH. On the left a scatter plot of the results obtained for 4–5 experiments is presented. The results are expressed as the percentage of the maximal ratio of immunoprecipitated CUL3/CSN4 (C) or CUL3/CAND1 (D) normalized on the CUL3/GAPDH ratio observed on the input gel. This ratio is significantly higher for CUL3-WT compared to CUL3-∆9 for both CSN4 and CAND1, independently of the neddylation state. Statistical comparisons used the Kruskal–Wallis test (* = *p* < 0.05 and ** = *p* < 0.01).

**Figure 5 ijms-23-05151-f005:**
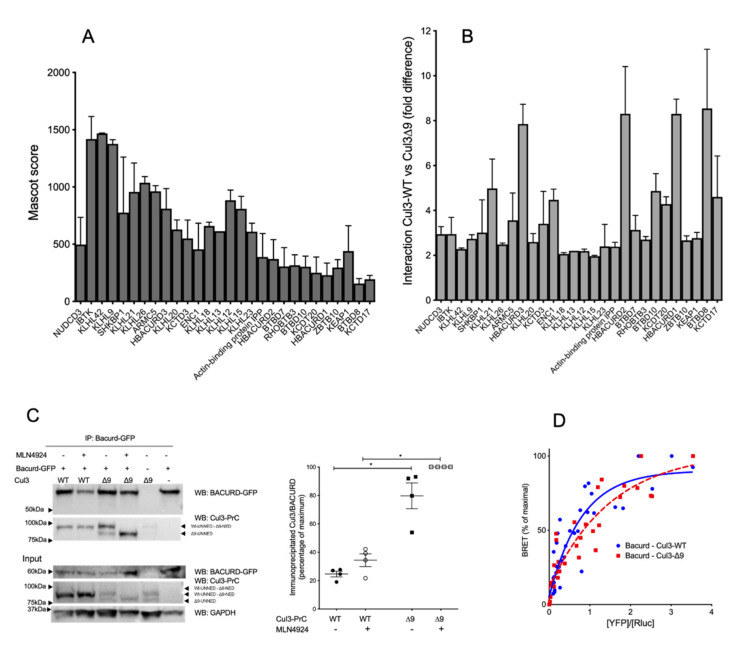
CUL3-WT and CUL3-∆9 interactions with BTB-proteins. Upper part: (**A**). and (**B**). interactions of BTB proteins with CUL3-WT and CUL3-∆9 analyzed by MS/MS. PrC-CUL3-WT and PrC-CUL3-∆9 were immunoprecipitated with HPC4 antibody, and the immunoprecipitate was analyzed by mass spectrometry. The abundance, as assessed by the Mascot score (**A**) and the difference in the interaction of BTB-proteins (**B**) with PrC-CUL3-WT and PrC-CUL3-∆9 are shown. Lower part: interactions of Bacurd with CUL3-WT and CUL3-∆9 analyzed by co-immunoprecipitation and BRET. (**C**). Bacurd-GFP was transfected in PrC-CUL3-WT or PrC-CUL3-∆9 inducible cell lines (see Materials and methods). Cells were induced (or not) with tetracycline (10 µg/mL) 32 h post-transfection and treated (or not) with MLN4924 (1µM) (Millenium Pharmaceuticals) 42 h post-transfection. Co-immunoprecipitation of Bacurd-GFP was performed by GFP-Trap followed by SDS-PAGE and transfer to nitrocellulose. On the left, a representative gel of the co-immunoprecipitation experiment blotted with the HPC4 (CUL3) and GFP (Bacurd) antibodies is shown. On the bottom, a Western blot of the same cellular extracts for Bacurd, CUL3, and GAPDH is shown. On the middle, a scatter plot of the results obtained for three experiments is presented. The results are expressed as the percentage of the maximal ratio of immunoprecipitated CUL3/Bacurd normalized on the CUL3/GAPDH ratio observed on the input gel. Statistical comparisons used the Kruskal–Wallis test on CUL3/Bacurd ratios (* = *p* < 0.05). (**D**). The interaction of nanoluc-Bacurd with YFP-CUL3-WT and YFP-CUL3-∆9 was analyzed by BRET experiments, as shown in Figure 2. Similar hyperbolic curves were obtained for CUL3-WT and CUL3-∆9, with non-significant different BRET_50_ (0.71 ± 0.16 and 1.01 ± 0.2, respectively, *n* = 3, *p* = 0.304). Statistical comparisons used the Student’s *t* test.

**Figure 6 ijms-23-05151-f006:**
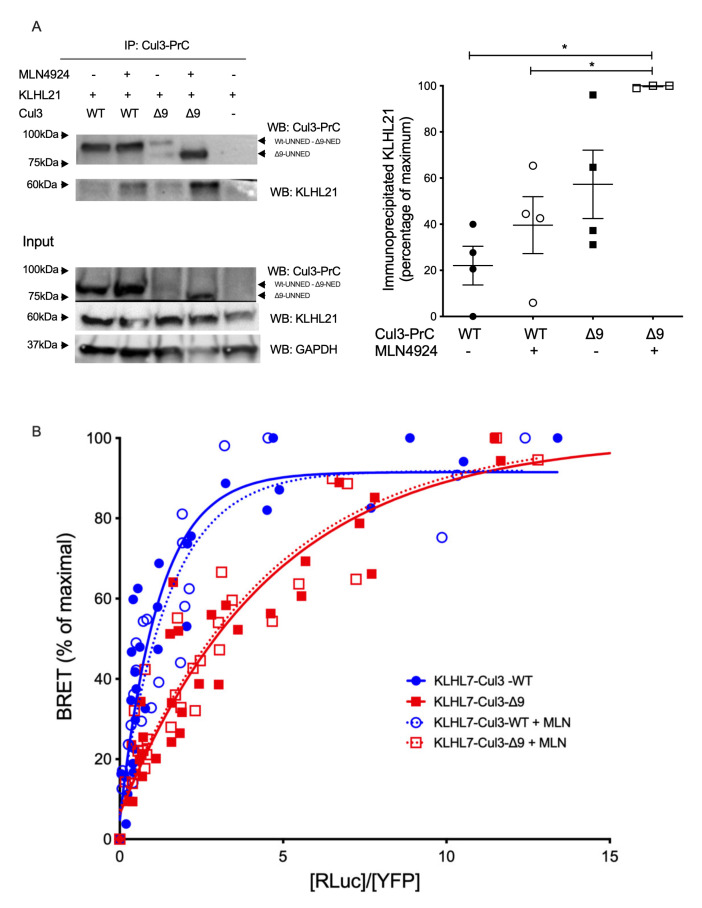
Interactions of other BTB-proteins (KLHL7 and KLHL21) with CUL3-WT and CUL3-∆9. (**A**). Flp-In^TM^ T-REx^TM^ 293 cells (Invitrogen), stably transfected with PrC-CUL3-WT or PrC-CUL3-∆9, were induced (or not) with tetracycline (10 µg/mL) for 16 h and treated (or not) with MLN4924 (1 µM) for 5 h. Co-immunoprecipitation of PrC-CUL3 was performed using HPC4-agarose beads (Roche) followed by SDS-PAGE and transfer to nitrocellulose. The left shows a representative gel of the co-immunoprecipitation experiment blotted with the polyclonal HPC4 (cell signaling) and KLHL21 (Invitrogen) antibodies. On the bottom, a Western blot of the same cellular extracts for CUL3, KLHL21, and GAPDH is shown. On the left, a scatter plot of the results obtained for three experiments is presented. The results are expressed as the percentage of the maximal ratio of immunoprecipitated KLHL21/CUL3. Statistical comparisons used the Kruskal–Wallis test on KLHL21/CUL3 ratios (* = *p* < 0.05). (**B**). The interaction of Luc-KLHL7 with YFP-CUL3-WT and YFP-CUL3-∆9 was analyzed by BRET experiments, as shown in Figure 2. Similar hyperbolic curves were obtained with CUL3-WT and CUL3-∆9 with non-significant different BRET_50_ (1.07 ± 0.37 and 2.71 ± 0.89, respectively, *n* = 3, *p* = 0.195). In addition, interestingly, deneddylation of CUL3-WT or CUL-∆9 with MLN4924 did not change these curves at all (BRET_50_ 1.27 ± 0.45 and 2.41 ± 0.66, respectively, *n* = 3, *p* = 0.226). Statistical comparisons used the Student’s *t* test.

**Figure 7 ijms-23-05151-f007:**
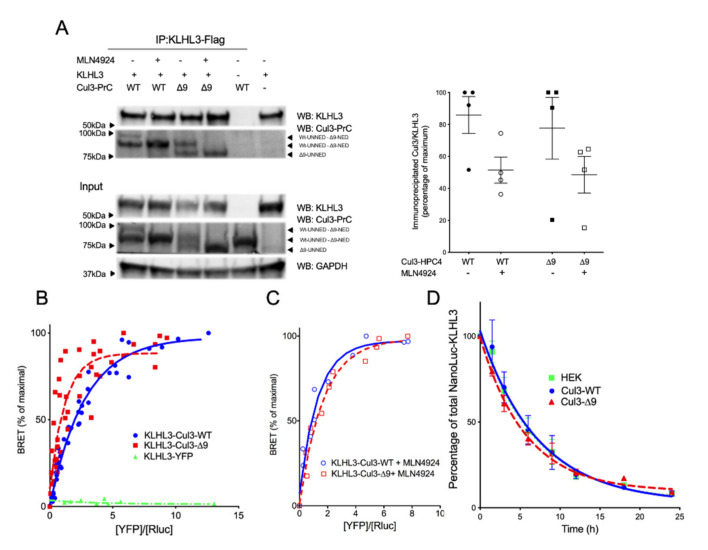
Dimerization, interaction and degradation of KLHL3. (**A**.) Interactions of CUL3-WT and CUL3-∆9 with KLHL3 Flag-KLHL3 was transfected in PrC-CUL3-WT or PrC-CUL3-∆9-inducible cell lines (see Materials and Methods), induced (or not) with tetracycline (10 µg/mL) for 16 h and treated (or not) with MLN4924 (1 µM) for 5 h. Co-immunoprecipitation of KLHL3 was performed using M2 Flag antibodies followed by SDS-PAGE and transfer to nitrocellulose. The left shows a representative gel of the co-immunoprecipitation experiment blotted with the polyclonal HPC4 (Cell signaling) and Flag-KLHL3 (Sigma) antibodies. The bottom shows a Western blot of the same cellular extracts for CUL3, KLHL3, and GAPDH. On the left, a scatter plot of the results obtained for four experiments is presented. The results are expressed as the percentage of the maximal ratio of immunoprecipitated CUL3/KLHL3 normalized on the CUL3/GAPDH ratio observed on the input gel. Statistical comparisons used the Kruskal–Wallis test on CUL3/KLHL3 ratios and were not significant. (**B**). The interaction of Luc-KLHL3 with YFP-CUL3-WT and YFP-CUL3-∆9 was analyzed with BRET experiments, as shown in Figure 2. Both CUL3-WT and CUL3-∆9 strongly interacted with KLHL3 in BRET experiments, but the BRET_50_ was significantly lower for CUL3-∆9 (0.9115 ± 0.33) as compared to CUL3-WT(1.855 ± 0.2) (*p* = 0.0478, *n* = 3). Statistical comparisons used the Student *t* test. (**C**). This twofold difference in the BRET_50_ is abolished by the pretreatment of the cells with MLN4924, the inhibitor of neddylation (BRET50 = 0.9143 and 1.172 for CUL3-WT and CUL3-∆9, respectively). (**D**). For KLHL3 degradation, Flp-InTM T-RExTM 293 cells stably transfected with PrC-CUL3-WT or PrC-CUL3-∆9 were transfected with Nanoluc-KLHL3. Twenty-four hours post-transfection, cells were treated with tetracycline (10 ng/mL) and 12 h later with cycloheximide and collected at different time points. Luminescence of KLHL3 was measured using a Mithras LB940 plate reader. Results are expressed as the percentage of the initial luminescent signal as a function of time (h). Similar decreasing curves were observed for KLHL3 in the presence of CUL3-WT or CUL3-∆9 with a half-life of 4.24 ± 0.75 h and 4.55 ± 0.58, respectively (*n* = 4; *p* = 0.7521). The Student *t* test was used for statistical comparisons.

**Figure 8 ijms-23-05151-f008:**
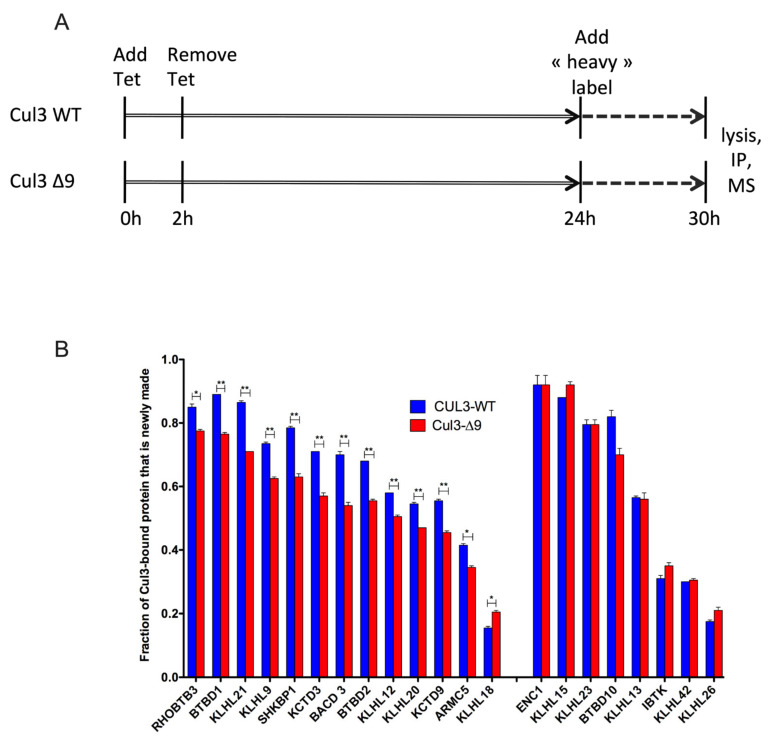
Cycling of CRL3-WT and CRL3-∆9 complexes analyzed by a SILAC experiment. (**A**). Flp-In^TM^ T-REx^TM^ 293 cells stably expressing PrC-CUL3-WT or PrC-CUL3-∆9 were transiently (2 h) induced with tetracycline in a normal “light” medium for 24 h, before switching them to a medium formulated with isotopically heavy lysine and arginine (“heavy” medium). After 6 h of incubation in “heavy” medium cells were lysed, prC-CUL3 was immunoprecipitated and the immunocomplexes were analyzed by SILAC quantitative mass spectrometry. (**B**). Twenty-one BTB-domain proteins (previously shown as differentially interacting with CUL3-WT and CUL3-∆9 in our label-free experiments) presented a newly made fraction > 0.2 in SILAC experiments. Among those 21 BTB adaptors, 12 had a statistically lower “newly synthetized” fraction in CUL3-∆9 complexes compared to CUL3-WT ones (left). Eight BTB adaptors presented no statistical difference in the distribution of “newly synthesized” fraction in CUL3-∆9 or CUL3-WT complexes (right). Only one presented a higher newly synthetized fraction in CUL3-∆9 complexes (middle). These results suggest that CUL3-∆9 is less performing in adaptor exchange and could inhibit CUL3-WT normal function by sequestering some abundant BTB adaptors, thus disturbing their “free” pool inside the cell. ** *p* < 0.05, * *p* < 0.01

**Figure 9 ijms-23-05151-f009:**
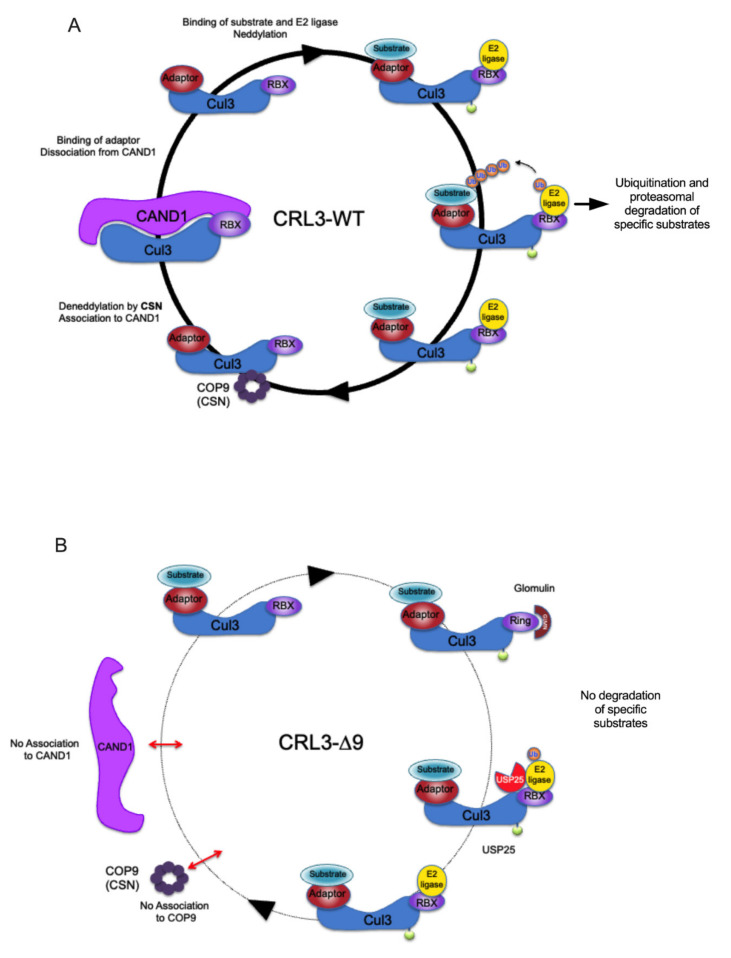
Hypothetical model of the functional defects of CUL3-∆9 CRL complex. (**A**). Classical model of CRL3 cycling. The different partners of the CRL3 cycle are indicated. The Cul3-Rbx1/CAND1 complex is inactive (left). Its dissociation allows the binding of adaptor and substrate and the neddylation of CUL3. This potentially active complex binds an E2 ligase and ubiquitinates the substrate, leading to the degradation by the proteasome. After substrate ubiquitination, the CRL3 complex interacts with the COP9 signalosome, resulting in the deneddylation of CUL3. The dissociation from the adaptor allows the re-association with CAND1. The different partners of the CRL cycle are indicated. (**B**). In the presence of CUL3-∆9, the CRL3 is unable to bind to COP9 (CSN4) and therefore to be deneddylated and to bind to CAND1. This interrupts the CRL3 cycle, which is essential for the recruitment of correct adaptors and substrates. In addition, the association of CUL3-∆9 to glomulin and USP25 could produce neddylated but potentially inhibited CRL3 complexes.

**Table 1 ijms-23-05151-t001:** BTB-domain proteins interact more with CUL3-∆9.

	Peptides	Score	Anova	Fold
**IBTK**	**32/26**	**1615/1221**	**0.02/0.01**	**2.2/3.69**
**KBTBD6**	**31/**	**1585/**	**0.05/**	**1.39/**
**KLHL42**	**26/22**	**1462/1472**	**0.000709/0.02**	**2.23/2.33**
**KLHL9**	**29/23**	**1413/1340**	**0.01/0.00825**	**2.92/2.55**
**SHKBP1**	**23/6**	**1261/289**	**0.000149/0.11**	**4.47/1.55**
**KLHL22**	**24/15**	**1238/770**	**0.03/0.65**	**1.67/1.06**
**KLHL21**	**21/13**	**1209/704**	**0.03/0.01**	**6.29/3.67**
**KLHL26**	**18/14**	**1091/983**	**0.0047/0.00809**	**2.42/2.55**
**KLHL8**	**21/11**	**1041/582**	**0.43/0.06**	**1.14/1.36**
**KCTD9**	**13/**	**1024/**	**0.13/**	**2.11/**
**BACD3**	**16/12**	**986/635**	**0.02/0.05**	**6.97/8.73**
**KLHL15**	**16/13**	**918/700**	**0.00887/0.03**	**1.92/2**
**BTBD9**	**18/11**	**886/524**	**0.38/0.19**	**1.08/1.72**
**KLHL12**	**13/17**	**796/972**	**0.07/0.0065**	**2.1/2.28**
**KLHL20**	**15/12**	**712/545**	**0.02/0.01**	**2.22/2.97**
**KCTD3**	**14/7**	**711/389**	**0.01/0.02**	**4.85/1.96**
**KLHL23**	**14/9**	**684/535**	**0.13/0.01**	**1.41/3.38**
**ENC1**	**13/3**	**683/228**	**0.00265/0.03**	**3.99/4.95**
**KLHL7**	**13/3**	**649/168**	**0.07/0.19**	**1.31/1.34**
**BTBD2**	**8/11**	**627/742**	**0.19/0.15**	**1.8/2.58**
**KLHL18**	**12/9**	**627/693**	**0.01/0.36**	**2/2.12**
**KLHL13**	**12/**	**614/**	**0.02/**	**2.2/**
**ACTIN-BINDING PROTEIN IPP**	**12/4**	**594/183**	**0.05/0.000248**	**2.19/2.58**
**KBTBD7**	**10/21**	**553/1281**	**0.1/0.36**	**1.37/1.09**
**BACD2**	**8/6**	**540/201**	**0.02/0.000164**	**10.41/6.2**
**RCBTB1**	**9/5**	**524/214**	**0.83/0.14**	**1.13/1.1**
**RHOBTB1**	**12/4**	**521/194**	**0.2/0.32**	**1.78/1.31**
**KBTBD4**	**8/5**	**501/208**	**0.1/0.67**	**1.43/1.77**
**BTBD7**	**11/2**	**470/139**	**0.03/0.09**	**3.78/2.48**
**KLHL36**	**9/2**	**455/97**	**0.18/0.49**	**1.15/2.28**
**KLHL25**	**10/5**	**453/318**	**0.37/0.00372**	**1.31/1.4**
**RHOBTB3**	**10/4**	**407/224**	**0.000957/0.1**	**2.84/2.56**
KBTBD2	7/	402/	0.63/	1.14/
**BTBD10**	**6/5**	**396/211**	**0.04/0.01**	**5.64/4.09**
**KCTD20**	**11/4**	**389/112**	**0.02/0.00129**	**3.95/4.61**
KLHL17	10/	376/	0.38/	1.17/
**KBTBD8**	**10/**	**369/**	**0.00828/**	**4.39/**
**BACD1**	**5/4**	**335/123**	**0.00486/0.000495**	**8.96/7.64**
KCTD18	7/9	302/312	0.03/0.58	1.44/1.09
RCBTB2	5/**3**	259/**115**	0.65/**0.03**	1.21/1.56
GIGAXONIN	7/	250/	0.09/	1.58/
BTBD1	5/9	229/558	0.32/0.06	2.73/2.66
**ZBTB10**	**6/8**	**226/365**	**0.01/0.000536**	**2.87/2.46**
**KEAP1**	**5/11**	**219/661**	**0.02/0.00048**	**3.02/2.51**
KLHL24	5/4	206/178	0.34/0.59	1.44/1.26
**BTBD8**	**5/3**	**200/112**	**0.03/0.00296**	**5.9/11.18**
KCTD6	4/	182/	0.64/	1.29/
KLHL28	5/2	172/123	0.2/0.71	1.3/1.19
**KCTD17**	**5**/4	**159**/228	**0.02**/0.16	**6.43**/2.77
KCTD2	2/2	118/156	0.08/0.21	6.44/3.19
ZNF131	3/	104/	0.59/	2.21/
KLHL11	3/	100/	0.25/	1.33/
KLHL5	4/2	82/136	0.24/0.17	1.49/1.69
**BTBD6**	**2/**	**76/**	**0.02/**	**8.91/**
KLHL2	2/	71/	0.76/	1.19/
ZBTB7A	/4	/197	/0.33	/2.21
KCTD5	/2	/98	/0.23	/2.75
**BACH2**	**/11**	**/538**	**/0.00639**	**/2.5**
**Galectin-3-binding protein**	**/4**	**/143**	**/0.04**	**/2.44**
**KAISO**	**/3**	**/127**	**/0.04**	**/3.13**
**BACH1**	**/15**	**/840**	**/0.01**	**/2.63**
**BTBD11**	**/10**	**/645**	**/0.01**	**/1.95**
ZBTB14	/4	/216	/0.11	/2.95
ZBTB21	/3	/83	/0.17	/1.82
ZBTB1	/2	/51	/0.12	/2.3
ZBTB17	/2	/56	/0.43	/1.74

Table presents a list of the BTB-proteins interacting with CUL3-∆9 than CUL3-WT. The number of different peptides for each protein (peptides, column 2), the Mascot score (score, column 3), the significance of the difference between CUL3-WT and CUL3-∆9 analyzed by Anova (Anova, column 4), and the fold increase in the CUL3-∆9 interaction versus the CUL3-WT interaction (fold, column 5) are indicated. The proteins in bold are those for which the interaction with CUL3-∆9 is significantly increased compared to CUL3-WT interaction in at least one experiment. These results synthesizes two different MS/MS experiments (experiment 1/experiment 2) in each column.

## Data Availability

Mass spectrometry raw data are available upon request on https://zenodo.org/record/5797037#.YcJm-RPMLow”, DOI 10.5281/zenodo.5797037 (accessed on 21 December 2021).

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
