# Peer review of "Cullin 3 Exon 9 Deletion in Familial Hyperkalemic Hypertension Impairs Cullin3-Ring-E3 Ligase (CRL3) Dynamic Regulation and Cycling"

_ijms, 2022, doi:10.3390/ijms23095151_

Round 1

Reviewer 1 Report

This manuscript highlights the importance of interacting partners of Cullin 3 in wild type and truncated forms observed in familial hyperkalemic hypertension (FHHt). The authors have used proteomics and BRET to test their hypothesis. Conceptually, the manuscript is well designed. The following comments are provided with the spirit of improving an already excellent paper. 

  1. Figure 2B. The banding pattern is very difficult to interpret, particularly for the CUL3-WT. There does not appear to be any evident NED form, but that may be due to clear over exposure of the blot. The MLN data does identify the main band as the UNNED form.  The CUL3D9 is easier to interpret because of the 57 aa deletion and the clear distinction of both forms.  Would a COP9 inhibitor help with interpretation? Either way, it would be helpful for the readers if you can label the NED and UNNED forms in each of the blots throughout the paper.
  2. Whenever possible, please label the Nedd and un-Nedd forms of Cul3 on each WB. Obviously, this will differ for WT and D9 versions. For example, this would be helpful in the input Cul3-PrC in Figure 4D.
  3. It would be most attractive is the western blots can be consistent, with the IP target on top and co-IP on the bottom or vice versa. This current is inconsistent as it switches back and forth (compare Figure 3C and 3D for example).
  4. Question about experimental design in Figure 2A. Some data are collected for 15 hours whereas others appear to be terminated after 5 hours. Would it be appropriate to convert the hyperbola into a straight-line double reciprocal line weaver-Burk plot.
  5. There are some linguistic errors in the manuscript. The authors are suggested to go through and correct errors in punctuation.

Author Response

The authors are grateful for the positive comments of the 2 reviewers and thank them for a careful analysis of this paper and their suggestions, which improve without any doubt the quality of the manuscript.

Reviewer 1

  1. Figure 2B. The banding pattern is very difficult to interpret, particularly for the CUL3-WT. There does not appear to be any evident NED form, but that may be due to clear over exposure of the blot. The MLN data does identify the main band as the UNNED form.  The CUL3D9 is easier to interpret because of the 57 aa deletion and the clear distinction of both forms.  Would a COP9 inhibitor help with interpretation? Either way, it would be helpful for the readers if you can label the NED and UNNED forms in each of the blots throughout the paper.

We agree that in Figure 1B the banding pattern was difficult to interpret. We have now repeated the experiment and loaded a reduced amount of protein. It is now easy to identify the NED form of CUL3-WT, which disappear after MLN treatment. We have labelled the NED and UNNED forms of CUL3-WT and CUL3-∆9 in this figure and the subsequent figures, where it is relevant. In addition, and according to referee 2’s requests, we have performed the mutation of the neddylation site (K712R) of CU3-WT and CUL3-∆9 and confirmed that the lower band correspond to the UNNED of CUL3-WT or -∆9 (see figure 3E).

  1. Whenever possible, please label the Nedd and un-Nedd forms of Cul3 on each WB. Obviously, this will differ for WT and D9 versions. For example, this would be helpful in the input Cul3-PrC in Figure 4D.

As mentioned in point 1 answer, it has been done, each time it was relevant.

  1. It would be most attractive is the western blots can be consistent, with the IP target on top and co-IP on the bottom or vice versa. This current is inconsistent as it switches back and forth (compare Figure 3C and 3D for example)

We understand the point. In the previous version of the manuscript, the proteins of the Western blots were ordered according to their molecular weights (the higher molecular weights at the top and the lower at the bottom) in the exact order of the native blot. We now have changed this and put the IP target at the top and the co-IP protein at the bottom for the immunoprecipitation experiment and ordered the input Western blot similarly.   

  1. Question about experimental design in Figure 2A. Some data are collected for 15 hours whereas others appear to be terminated after 5 hours. Would it be appropriate to convert the hyperbola into a straight-line double reciprocal line weaver-Burk plot.

We realize that we did not explain correctly our BRET experiments and it is done now pages 3, lines 142 to 145, and page 4, lines 190-196. In figure 2A, the BRET signal is not expressed in function of time but in function of the ratio between the amounts of the BRET donor (luciferase-protein) and the BRET acceptor (YFP-protein). Due to expression limitation, we were unable to obtain a ratio > 5 for CUL3- WT homodimers, whereas ratio up to 15 were obtained for the CUL3-∆9 homodimers or heterodimers. In order to clarify the Figure, we modified the relationship between BRET signal and [RLuc]/[YFP] ratio, to a ratio of 5 for all 3 dimers. According to numerous previous reports, this relationship is linear if there is a non-specific interaction between the donor and the acceptor and hyperbolic if there is a specific interaction between the 2 proteins (for more details, see the following review reference ;  Achour L, Kamal M, Jockers R, Marullo S. Using quantitative BRET to assess G protein-coupled receptor homo- and heterodimerization. Methods Mol Biol. 2011;756:183-200. doi: 10.1007/978-1-61779-160-4_9). According to the observed results, there is no doubt that CUL3-WT and CUL3-∆9 form homo- and heterodimers and that the curves are hyperbolic.

  1. There are some linguistic errors in the manuscript. The authors are suggested to go through and correct errors in punctuation.

The manuscript has been carefully verified for linguistic and punctuations errors and corrected by a native-english coauthor.

Reviewer 2 Report

The authors seek to understand the molecular mechanisms underlying how deletion of exon 9 in Cullin 3 causes familiar hyperkalemic hypertension by affecting the interaction of Cullin3 with various regulatory proteins, leading to aberrant neddylation, locked in its inactive complex form. The manuscript is clearly written. However, the following concerns should be addressed before its publication.

  1. Page 2 line 76-78, the sentence in description of how to organize the results section should be deleted.
  2. Figure 1B: the authors should explain of speculate why the expression of Cul3-delta 9 is less than that of WT?
  3. Figure 2B, if half-life is comparable, the authors should explore or discuss if the mRNA half-life reduction or reduced translation might cause the reduced expression of Cul3-delta 9.
  4. Figure 3D, the authors should show if mutating the key neddylation Lys residues can block interaction of Cul3-delta 9 with USP25
  5. Figure 4C: does it mean CSN4 binding motif localizes in the exon 9 coding (aa 403-459) region?
  6. Figure 5, given the critical role of KLHL3 in familiar hyperkalemic hypertension , it will be nice for the authors to explore if deletion of exon 9 affect Cullin 3 interaction with KLHL3 and it ability in degrading WINK1.

Author Response

The authors are grateful for the positive comments of the 2 reviewers and thank them for a careful analysis of this paper and their suggestions, which improve without any doubt the quality of the manuscript.

Page 2 line 76-78, the sentence in description of how to organize the results section should be deleted.

Sorry for this mistake, which has been corrected in the new version.

Figure 1B: the authors should explain or speculate why the expression of Cul3-delta 9 is less than that of WT?

Figure 2B, if half-life is comparable, the authors should explore or discuss if the mRNA half-life reduction or reduced translation might cause the reduced expression of Cul3-delta 9.

These two points raised by the reviewer deal with the same question: why the 2 stable Flp-InTM T-RExTM 293 cells expressing CUL3-WT or CUL3-∆9 under the same inducible promoter in the same locus produce different amounts of the protein? To answer this question, we first quantified the amount of the proteins in the 2 cell lines on several Western blots and identified a significant but moderate difference (26.4%±6.4 p=0.0033). (Supplemental Figure A). According to Supplemental Figure 1B, this difference was not due to protein degradation. We then analyzed the levels of CUL3 mRNA in the 2 cell lines by RT-qPCR. A non-significant reduction (9%) was observed for CUL3-∆9 compared to CUL-WT mRNA (Suppl. Figure 1C). Finally, the mRNA degradation was not different in the 2 cell lines (Suppl. Figure 1D). These elements are now discussed in the manuscript (page 2 and 3 lines 88-99) and we speculate that the possible defect occurs at the translation level.

Figure 3D, the authors should show if mutating the key neddylation Lys residues can block interaction of Cul3-delta 9 with USP25

We followed the suggestion of the reviewer and mutated the unique Lys, which is neddylated In CUL3, for Arg (K712R). Western blots and co-immunoprecipitations (Figure 3E) confirm i) that the upper band is indeed the neddylated form of Cul3, since this band disappears after K712R mutation, ii) that CUL3-∆9 interacts more with USP25 than CUL3-WT, but this interaction is strongly reduced by the K712R mutation, confirming that this interaction is dependent upon CUL3-∆9 neddylation.

Figure 4C: does it mean CSN4 binding motif localizes in the exon 9 coding (aa 403-459) region?

Figure 4C indicates that the COP9 signalosome does not interact with CUL3-∆9. As discussed in the manuscript (page 17 lines 583-587), a previous study has delineated the binding site of CUL3 to COP9 signalosome as the domain including the amino-acids 461 to 586 (Cornelius RJ et al Am J Physiol Renal Physiol 2018, 315: F1006-18). This domain is upstream of exon 9 (amino acids 403-459), indicating that the absence of binding of CUL3-∆9 to CSN is more the consequence of a folding change of the protein, rather the direct deletion of the binding site.

Figure 5, given the critical role of KLHL3 in familiar hyperkalemic hypertension , it will be nice for the authors to explore if deletion of exon 9 affect Cullin 3 interaction with KLHL3 and it ability in degrading WINK1.

We are sorry that the supplementary Figure 1 was apparently not communicated to the reviewers. As described in this figure and in the text (page 13 lines 409-424), the interaction of CUL3-WT and CUL3-∆9 was explored by both co-immunoprecipitation and BRET. These experiments indicate that there is no difference in interaction of CUL3-WT and CUL3∆9 with KLHL3. This figure is now integrated in the manuscript as figure 7. In addition, we studied the degradation of WNK4 by Western blot in presence or not of KLHL3 and either USP25, glomulin or CUL3-∆9. The results show, as expected, that KLHL3 dramatically increases the degradation of WNK4 and CUL3-∆9 reduces this degradation. More unexpected is the major inhibitory effect of USP25 on the degradation of WNK4, whereas USP28, a closely related USP which does not interact with CUL3, is without effect. We did not observe also any effect of glomulin. These results are now presented in Supplementary figure 2.

Round 2

Reviewer 2 Report

The authors have addressed most of the raised concerns.